# Identification of five sleep-biopsychosocial profiles with specific neural signatures linking sleep variability with health, cognition, and lifestyle factors

Aurore A. Perrault[1,2,3☯*], Valeria Kebets[4,5,6,7,8☯*], Nicole M. Y. Kuek[4,5,6], Nathan E. Cross[1,2,9], Rackeb Tesfaye[8], Florence B. Pomares[1,2], Jingwei Li[4,5,6,10,11], Michael W. L. Chee[5], Thien Thanh Dang-Vu[1,2], B. T. Thomas Yeo[4,5,6,12,13,14]

**1** Sleep, Cognition and Neuroimaging Lab, Department of Health, Kinesiology and Applied Physiology & Center for Studies in Behavioral Neurobiology, Concordia University, Montreal, Canada, **2** Centre de Recherche de l'Institut Universitaire de Gériatrie de Montréal, CIUSSS Centre-Sud-de-l'Ile-de-Montréal, Montreal, Canada, **3** Sleep & Circadian Research Group, Woolcock Institute of Medical Research, Macquarie University, Sydney, Australia, **4** Department of Electrical and Computer Engineering, National University of Singapore, Singapore, Singapore, **5** Centre for Sleep and Cognition & Centre for Translational Magnetic Resonance Research, Yong Loo Lin School of Medicine, National University of Singapore, Singapore, Singapore, **6** N.1 Institute for Health, National University of Singapore, Singapore, Singapore, **7** McConnell Brain Imaging Centre (BIC), Montreal Neurological Institute (MNI), McGill University, Montreal, Canada, **8** McGill University, Montreal, Canada, **9** School of Psychology, University of Sydney, Sydney, Australia, **10** Institute of Neuroscience and Medicine (INM-7: Brain and Behavior), Research Center Jülich, Jülich, Germany, **11** Institute of Systems Neuroscience, Medical Faculty and University Hospital Düsseldorf, Heinrich Heine University Düsseldorf, Düsseldorf, Germany, **12** Department of Medicine, Human Potential Translational Research Programme & Institute for Digital Medicine (WisDM), Yong Loo Lin School of Medicine, National University of Singapore, Singapore, Singapore, **13** Integrative Sciences and Engineering Programme (ISEP), National University of Singapore, Singapore, Singapore, **14** Martinos Center for Biomedical Imaging, Massachusetts General Hospital, Charlestown, Massachusetts, United States of America

☯ These authors contributed equally to this work.
* aurore.perrault@gmail.com (AAP); valkebets@gmail.com (VK)

## Abstract

Sleep is essential for optimal functioning and health. Interconnected to multiple biological, psychological, and socio-environmental factors (i.e., biopsychosocial factors), the multidimensional nature of sleep is rarely capitalized on in research. Here, we deployed a data-driven approach to identify sleep-biopsychosocial profiles that linked self-reported sleep patterns to inter-individual variability in health, cognition, and lifestyle factors in 770 healthy young adults. We uncovered five profiles, including two profiles reflecting general psychopathology associated with either reports of general poor sleep or an absence of sleep complaints (i.e., sleep resilience), respectively. The three other profiles were driven by the use of sleep aids and social satisfaction, sleep duration, and cognitive performance, and sleep disturbance linked to cognition and mental health. Furthermore, identified sleep-biopsychosocial profiles displayed unique patterns of brain network organization. In particular, somatomotor network connectivity alterations were involved in the relationships between sleep and

which permits unrestricted use, distribution, and reproduction in any medium, provided the original author and source are credited.

**Data availability statement:** Data from the HCP dataset is publicly available (https://www.humanconnectome.org/). Data points underlying Figs 2A, 3A, 4A, 5A, 6A, and 7A as well as Table A and Figs A, D, and G in S1 Text, as well as the list of participants who passed our MRI quality control are presented in S1 Data. The brain parcellation can be obtained at (https://github.com/ThomasYeoLab/CBIG/tree/master/stable_projects/brain_parcellation/Schaefer2018_LocalGlobal), while the code for the CCA and GLM analyses can be found at (https://github.com/valkebets/sleep_biopsychosocial_profiles) and on Zenodo (DOI: 10.5281/zenodo.16624810). Chord diagrams were generated using previously published code (https://github.com/ThomasYeoLab/CBIG/tree/master/stable_projects/predict_phenotypes/ChenTam2022_TRBPC/figure_utilities/chord).

**Funding:** The author(s) received no specific funding for this work.

**Competing interests:** The authors have declared that no competing interests exist.

**Abbreviations:** BMI, body mass index; CCA, canonical correlation analysis; DAN, dorsal attention networks; DMN, default mode network; FD, framewise displacement; fMRI, functional MRI;GLM, generalized linear models; GSR, global signal regression; HCP, Human Connectome Project; LCs, latent components; MRI, magnetic resonance imaging; PSQI, Pittsburgh Sleep Quality Index; RT, reaction times; RDoC, research domain criteria;RSFC, resting-state functional connectivity;SVD, singular value decomposition,TPN,temporoparietal network.

biopsychosocial factors. These profiles can potentially untangle the interplay between individuals' variability in sleep, health, cognition, and lifestyle—equipping research and clinical settings to better support individual's well-being.

## Introduction

Decades of research have established that sleep is interconnected to multiple biological, psychological, and socio-environmental factors (i.e., biopsychosocial factors) [1–4]. Importantly, sleep difficulties are among the most common comorbidities of mental and physical disorders [5–8], highlighting the central role of sleep in health. Despite the recognition that sleep is a unique marker for optimal health [9,10] and a potential transdiagnostic therapeutic target, its multidimensional and transdisciplinary nature is rarely capitalized on in research. Traditionally, single-association studies have investigated the relationship between a single dimension of sleep (e.g., duration, quality, and onset latency) and/or a single outcome of interest. Such unidimensional studies have independently linked insufficient or poor sleep to a wide range of negative outcomes separately, including cognitive difficulties [11,12], brain connectivity changes [13–17], decreased physical health [7,18], poor mental health and well-being [8,19], as well as increased risks for cardiovascular disease [7,20,21], neurodegenerative disease [22,23], and psychiatric disorders [8,24]. However, by treating sleep as a binary domain (e.g., good versus poor sleep, short versus long), these studies fail to capture the multidimensional nature of sleep and the multiple intricate links with biological, psychological, and socio-environmental (i.e., biopsychosocial) factors. Therefore, it remains unclear which biopsychosocial factors are most robustly associated with sleep traits and whether these factors are supported by similar neural mechanisms.

Adding to the complexity of these relationships is how sleep and good sleep health are defined. Characterizing sleep is a challenging task because of its multidimensional nature [25]. Sleep can be defined by its quantity (i.e., sleep duration) and quality (i.e., satisfaction, efficiency), as well as in terms of regularity, timing, and alertness [9]. These dimensions are deemed particularly relevant when defining sleep health [9], as they each have been related to biopsychosocial outcomes. Different sleep dimensions can also be described as either "good" or "bad" sleep, without necessarily affecting one another, e.g., short sleep duration is not systematically associated with poor sleep quality. Another important aspect of sleep is how it is subjectively characterized. For instance, our perception of sleep can influence daytime functioning [26] and can be ascribed to certain behaviors that differ from objective reports [27,28].

Reconciling the multiple components of sleep and the complex connections to a myriad of biopsychosocial factors requires frameworks grounded in a multidimensional approach. The biopsychosocial model has long been used to assert that biological (e.g., genetics and intermediate brain phenotypes), psychological (e.g., mood and behaviors), and social factors (e.g., social relationships, economic status) are all significant contributors to health and disease [2,3]. Indeed, the biopsychosocial

model has been used to establish current diagnostic and clinical guidelines, such as the World Health Organization's International Classification of Functioning, Disability and Health, and is considered central to person-centered care [29]. Hence, statistical methods that enable us to interrogate the complex interconnected relationships within and between sleep and biopsychosocial factors can advance our understanding of optimal health and functioning across the life span. Multivariate data-driven techniques can help disentangle these complex interrelations by deriving latent components that optimally relate multidimensional data sets in a single integrated analysis. A few studies have used such techniques to account for the multidimensional components of sleep and biopsychosocial factors separately [15,30–34]. However, no study has integrated both multidimensional components of sleep and biopsychosocial factors to derive profiles that can account for the dynamic interplay among biopsychosocial factors in adults and link such components with brain network organization.

Deploying multivariate data-driven techniques requires a large sample size to identify latent components (LCs) that can be generalized well [35–37]. One such optimal dataset is the Human Connectome Project dataset (HCP) [38], as it comprises a wide range of self-reported questionnaires about lifestyle, mental and physical health, person-ality, and affect, as well as objective measures of physical health and cognition from over a thousand healthy young adults. Moreover, the HCP dataset stands out as one of the rare large-scale datasets that implemented a detailed assessment of sleep health, i.e., the Pittsburgh Sleep Quality Index (PSQI) [39]. This standardized sleep ques-tionnaire, used both by clinicians and researchers, assesses different dimensions of sleep health in 19 individual items, creating 7 sub-components defining different dimensions of sleep, including sleep duration, satisfaction, and disturbances.

Beyond sleep-biopsychosocial profiling, the HCP dataset also provides the opportunity to explore the neural signatures of these sleep-biopsychosocial profiles using magnetic resonance imaging (MRI). Multiple studies have shown that neural signal fluctuation patterns during rest (i.e., resting-state functional connectivity; RSFC) are sensitive to sleep dimensions (e.g., sleep duration, sleep quality) [14,15,17,34,40], but also predictive of psychopathology (e.g., depressive symptoms, impulsivity) [41,42] and cognitive performance [14,40]. However, the way large-scale network organization may differen-tially affect individuals' variability in sleep, psychopathology, cognition, and lifestyle, remains to be characterized beyond unidimensional association studies. Such holistic biopsychosocial approaches are not only in line with established diag-nostic frameworks but also with initiatives such as the NIMH's Research Domain Criteria (RDoC) that encourage inves-tigating mental disorders as continuous dimensions rather than distinct categories by integrating data from genomics, neural circuitry, and behavior [43–45].

Identifying vulnerability markers constitutes a first step towards forecasting disease trajectories and designing multi-modal multidimensional targeted therapies. Given the increasing recognition that sleep has a central role in health and well-being, we believe that sleep profiles should be included as a core aspect of these markers. Hence, in this study, we sought to take a multidimensional data-driven approach to identify sleep-biopsychosocial profiles that simultaneously relate self-reported sleep patterns to biopsychosocial factors of health, cognition, and lifestyle in the HCP cohort of healthy young adults [38]. We further explored patterns of brain network organization associated with each profile to better under-stand their neurobiological underpinnings.

## Results

We applied canonical correlation analysis (CCA) to derive latent components (LCs) linking the 7 sub-components of the PSQI to 118 biopsychosocial measures (spanning cognitive performance, physical and mental health, personality traits, affects, substance use, and demographics; Table A in S1 Text) in 770 healthy adults from the S1200 release of the HCP dataset [38] (Fig 1A). Participants were young adults between 22 and 36 years old (mean 28.86 ± 3.61 years old, 53.76% female), were generally employed full-time (70.7%) and were mostly white (78%; see Table 1 for Demographics).

**A.** Canonical correlation analysis (CCA)

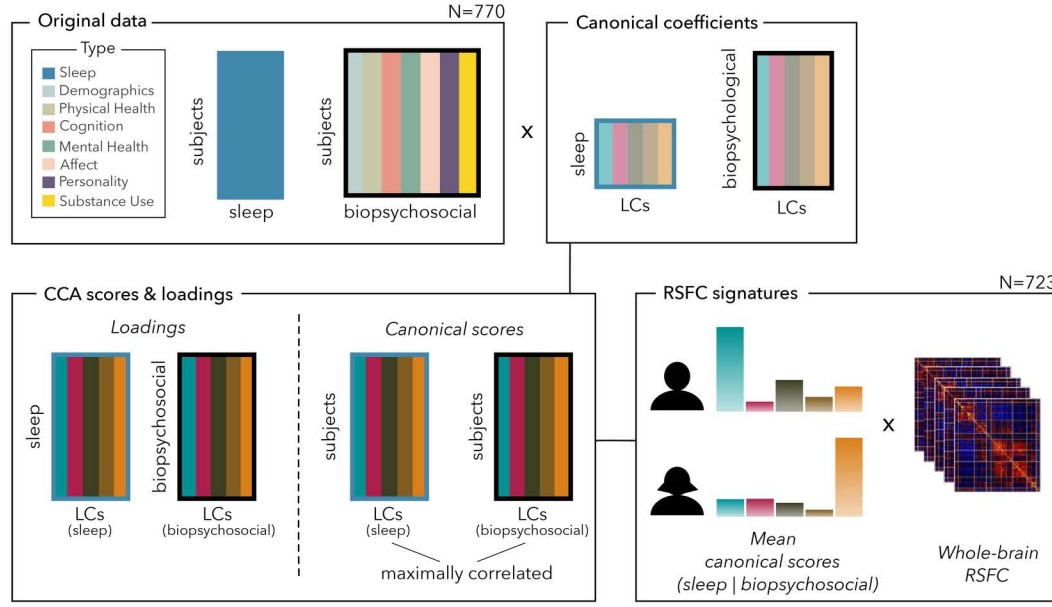

**B.** Sleep-biopsychosocial profiles (LCs)

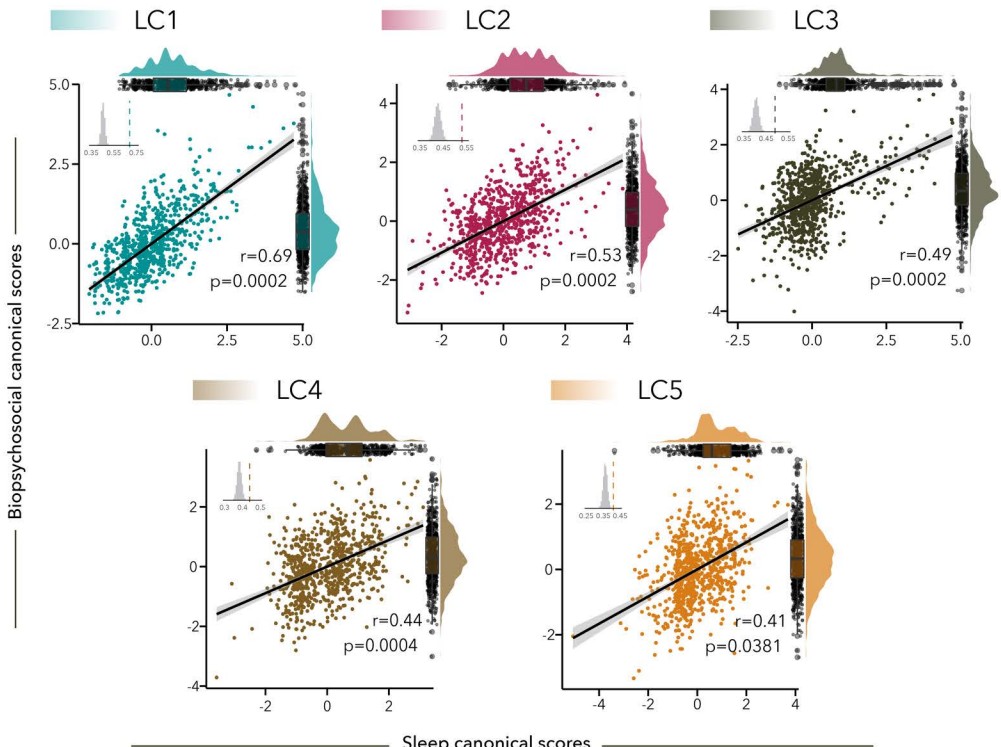

Figure 1

**Fig 1. Canonical correlation analysis reveals five sleep-biopsychosocial profiles (LCs). (A)** Canonical correlation analysis (CCA) flowchart and RSFC signatures; **(B)** Scatter plots showing correlations between biopsychosocial and sleep canonical scores. Each dot represents a different participant. The inset shows the null distribution of canonical correlations obtained by permutation testing; note that the null distribution is not centered at zero. The dashed line indicates the actual canonical correlation computed for each LC. The distribution of sleep (top) and biopsychosocial (right) canonical scores is shown on rain cloud plots.

**Table 1. Demographics (*N*=770).**

| Characteristics | *N*=770 |
|---|---|
| Biological sex (*n* \| %) | |
| Female | 414 \| 53.76% |
| Male | 356 \| 46.23% |
| Age (years) | |
| Mean±SD | 28.86 ±3.61 |
| Range | [22–36] |
| Education (years) | |
| Mean±SD | 15.02±1.73 |
| Range | [11–17] |
| Race (*n* \| %) | |
| Am. Indian/Alaskan Nat. | 2 \| 0.25% |
| Asian/Nat. Hawaiian/Other Pacific Is. | 42 \| 5.45% |
| Black or African Am. | 90 \| 11.68% |
| More than one | 19 \| 2.46% |
| Unknown or not reported | 16 \| 2.07% |
| White | 601 \| 78.05% |
| Ethnicity (*n* \| %) | |
| Hispanic/Latino | 78 \| 10.12% |
| Not Hispanic/Latino | 684 \| 88.83% |
| Unknown or not reported | 8 \| 1.03% |
| Employment status (*n* \| %) | |
| Full-time | 545 \| 70.77% |
| Part-time | 132 \| 17.14% |
| Not working | 93 \| 12.07% |
| School status (*n* \| %) | |
| In school | 158 \| 20.51% |
| Not in school | 612 \| 79.48% |
| Yearly income (*n* \| %) | |
| <10,000 US$ | 50 \| 6.49% |
| 10,000–20,000 US$ | 50 \| 6.49% |
| 20,000–30,000 US$ | 94 \| 12.20% |
| 30,000–40,000 US$ | 101 \| 13.11% |
| 40,000–50,000 US$ | 76 \| 9.87% |
| 50,000–75,000 US$ | 165 \| 21.42% |
| 75,000–100,000 US$ | 112 \| 14.54% |
| >100,00 US$ | 122 \| 15.84% |
| Relationship status (*n* \| %) | |
| In a relationship | 363 \| 47.14% |
| Not in a relationship | 407 \| 52.85% |
| PSQI total score | |
| Mean±SD | 5.14 ±2.17 |
| Range | [0–19] |

PLOS Biology

## Five LCs linking sleep and biopsychosocial factors

Out of the seven significant LCs that were derived, five LCs delineating multivariate relationships between sleep and biopsychosocial factors were supported by current sleep literature (Fig 1B; a description of LC6 and LC7 can be found in the Supplementary Results A and Fig A in S1 Text). While LC1 and LC2 were defined by general patterns of sleep (either general poor sleep or sleep resilience), LCs 3–5 reflected more specific sub-components of the PSQI, all associated with specific patterns of biopsychosocial factors; they also showed to be less robust and generalizable than LCs 1–2, as they did not survive cross-validation in our control analyses. The 5 LCs respectively explained 88%, 4%, 3%, 2%, and 1% of the covariance between the sleep and biopsychosocial data. While LC1 accounted for a substantial amount of covariance between sleep and biopsychosocial measures, LCs 2–5 highlighted distinct covariance patterns that were driven by specific sleep dimensions, probably representative of only a fraction of the participants, or present in all or most participants but with less prominence.

LC1 was characterized by a general pattern of poor sleep, including decreased sleep satisfaction, longer time to fall asleep, greater complaints of sleep disturbances, and daytime impairment, as well as greater (i.e., worse) psychopathology (e.g., depression, anxiety, somatic complaints, and internalizing behavior) and negative affect (e.g., fear, anger, and stress—Fig 2A).

Similarly, LC2 was also driven by greater psychopathology, especially attentional problems (e.g., inattention, ADHD), low conscientiousness, and negative affect (Fig 3A). In terms of sleep, however, in contrast to the first LC, greater psychopathology was only related to higher complaints of daytime impairment without complaints of sleep difficulties, suggesting sleep resilience.

LC3 was mostly characterized by sleep aids intake (i.e., sleep medication PSQI sub-component) and, to a lesser extent, a lack of daytime functioning complaint. Surprisingly, LC3 was not driven by any attentional problem but was related to worse performance in visual episodic memory and emotional recognition. Moreover, sleep aids intake was mainly related to satisfaction in social relationships (Fig 4A).

While LC4 was solely driven by sleep duration (i.e., not sleeping enough—reporting <6–7 h/night), LC5 was mostly characterized by the presence of sleep disturbances that can encompass multiple awakenings, nocturia, and breathing issues, as well as pain or temperature imbalance. In LC4, short sleep duration was associated with worse accuracy and longer reaction time at multiple cognitive tasks tapping into emotional processing, delayed reward discounting, language, fluid intelligence, and social cognition. LC4 was also characterized by higher aggressive behavior and lower agreeableness (Fig 5A).

Interestingly, sleep disturbances in LC5 were also associated with aggressive behavior and worse cognitive performance (e.g., in language processing and working memory), but were mostly characterized by critical items on mental health assessments (i.e., anxiety, thought problems, and internalization) and substance abuse (i.e., alcohol and cigarette use—Fig 6A).

## Sleep and biopsychosocial profiles exhibit distinct signatures of resting-state brain connectivity

In terms of brain organization, the 5 LCs revealed distinct patterns of network connectivity. Specifically, we examined patterns of both within-network and between-network connectivity (see Fig B in S1 Text for subcortical-cortical patterns).

Greater (averaged) biopsychosocial and sleep composite scores on LC1 were associated with increased RSFC between subcortical areas and the somatomotor and dorsal attention networks (Fig 2B and 2C), and a decreased RSFC between the temporoparietal network (TPN) and these two networks. The visual network showed a flattened distribution of segregation/integration ratio (i.e., more variability in segregation and integration among the parcels of the network). The amygdala and nucleus accumbens exhibited asymmetrical patterns in the segregation/integration ratio with the left side being more segregated (Fig 2D). Meanwhile, LC2 was associated with increased RSFC between the dorsal attention and control network but decreased RSFC between dorsal attention and the temporoparietal and limbic networks (Fig 3B and

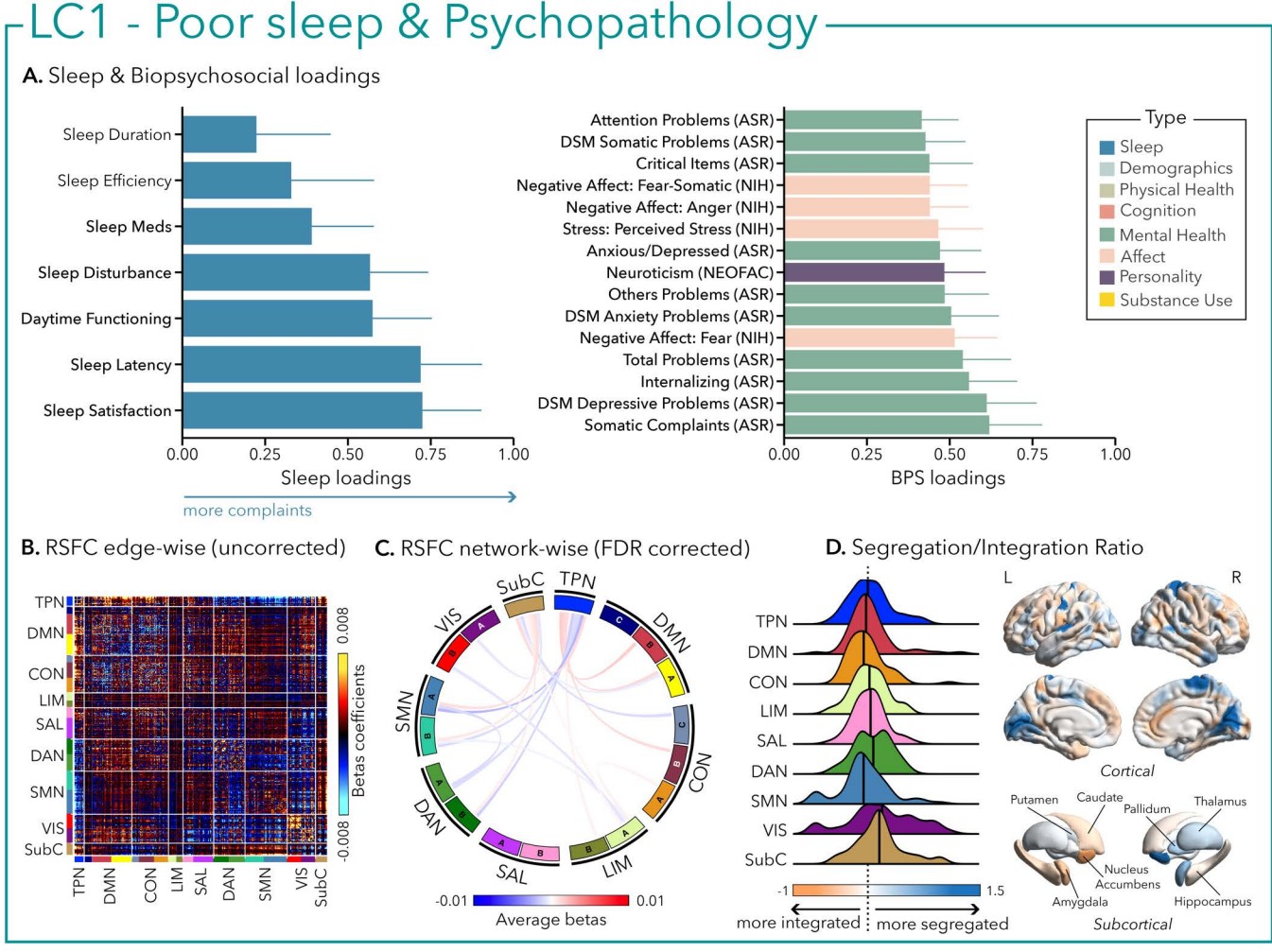

## LC1 - Poor sleep & Psychopathology

**A.** Sleep & Biopsychosocial loadings

**B.** RSFC edge-wise (uncorrected)

**C.** RSFC network-wise (FDR corrected)

**D.** Segregation/Integration Ratio

**Fig 2. The first latent component (LC1) reflects the association between poor sleep and psychopathology. (A)** Sleep loadings (left) and top 15 strongest biopsychosocial (BPS) loadings (right) for LC1. Greater loadings on LC1 were associated with higher measures of poor sleep and psychopathology. Higher values on sleep (blue) and biopsychosocial (green, purple, and pink) loadings indicate worse outcomes. Error bars indicate bootstrapped-estimated confidence intervals (i.e., standard deviation) and measures in bold indicate statistical significance (after FDR correction $q < 0.05$); **(B)** Unthresholded edge-wise beta coefficients obtained from generalized linear models (GLM) between participants' LC1 canonical scores (i.e., averaged sleep and biopsychosocial canonical scores) and their RSFC data; **(C)** FDR-corrected network-wise beta coefficients computed with GLMs within and between 17 large-scale brain networks [46] and subcortical regions [47]. **(D)** Distribution of the integration/segregation ratio in each of the 7 large-scale brain networks and subcortical regions associated with LC1 (left). The dashed line indicates the median of all parcels, and the bold black lines represent the median for each network. The integration/segregation ratio values for the 400 Schaeffer parcellation [48] and 7 subcortical regions are projected on cortical and subcortical surfaces (right). See S1 Data for underlying data.

3C), a higher segregation of nodes within the TPN and increased integration within the right thalamus (Fig 3D). Higher composite scores in LC3 were associated with increased RSFC within the visual and default mode networks (Fig 4B and 4C). The segregation/integration ratio within the default mode exhibited a flattened distribution (i.e., high variability in segregation and integration among parcels), but there was an increased segregation in the limbic and visual networks (Fig 4D). While greater composite scores in LC4 were associated with widespread patterns of hypo- or hyper-connectivity within and between every network, the somatomotor network specifically exhibited an altered pattern of segregation and integration (Fig 5B–5D). Finally, we found that greater averaged composite scores in LC5 were mainly associated with

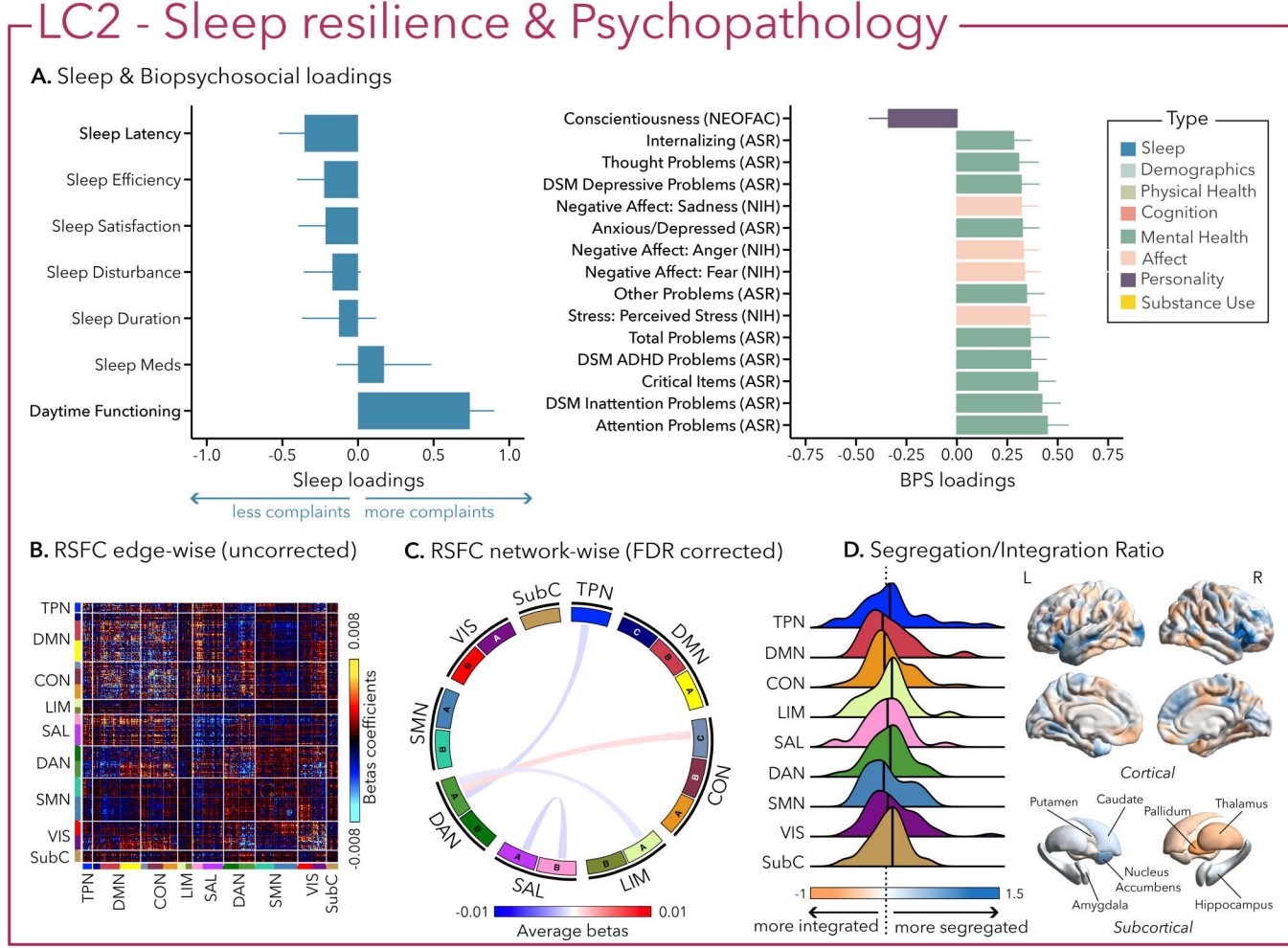

**Fig 3. The second latent component (LC2) reflects the association between sleep resilience and psychopathology. (A)** Sleep loadings (left) and top 15 strongest biopsychosocial (BPS) loadings (right) for LC2. Greater loadings on LC2 were associated with higher measures of complaints of daytime dysfunction and psychopathology. Positive values on sleep (blue) loadings indicate worse outcomes while positive values on biopsychosocial (green, purple, pink) loadings reflect higher magnitude on these measures. Error bars indicate bootstrapped-estimated confidence intervals (i.e., standard deviation) and measures in bold indicate statistical significance. **(B)** Unthresholded edge-wise beta coefficients obtained from generalized linear models (GLM) between participants' LC2 canonical scores (i.e., averaged sleep and biopsychosocial canonical scores) and their RSFC data; **(C)** FDR-corrected network-wise beta coefficients computed with GLMs within and between 17 large-scale brain networks [46] and subcortical regions [47]. **(D)** Distribution of the integration/segregation ratio in each of the 7 large-scale brain networks and subcortical regions associated with LC2 (left). The dashed line indicates the median of all parcels, and the bold black lines represent the median for each network. The integration/segregation ratio values for the 400 Schaeffer parcellation [48] and 7 subcortical regions are projected on cortical and subcortical surfaces (right). See S1 Data for underlying data.

reduced within-network connectivity in the somatomotor, dorsal, and ventral attention networks (Fig 6B and 6C) but no strong pattern of segregation/integration ratio change (Fig 6D).

## Post hoc associations with socio-demographics, health, and family history of mental health

We found a number of significant associations between LC composite scores and socio-economic (e.g., education level and household income) and socio-demographic factors (e.g., race, ethnicity; See Table D and Supplemental Results B in S1 Text)

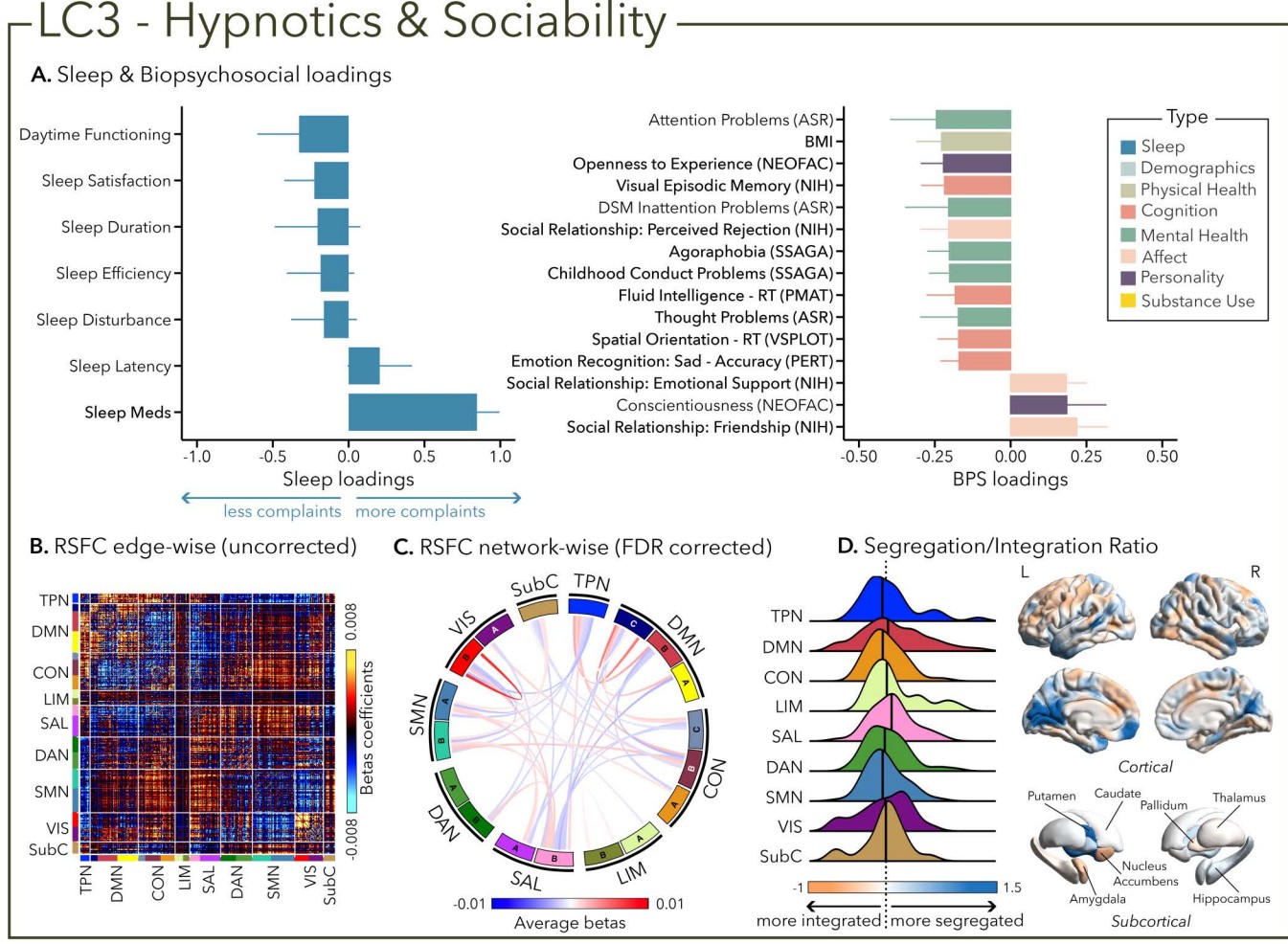

**Fig 4. The third latent component (LC3) reflects the association between sleep aids use and sociability. (A)** Sleep loadings (left) and top 15 strongest biopsychosocial (BPS) loadings (right) for LC3. Greater loadings on LC3 were associated with the use of sleep aids (hypnotics) and measures of positive social relationships, lower body mass index (BMI), and poor visual episodic memory performance. Positive values on sleep (blue) loadings indicate worse outcomes while positive values on the mental health (green), affect (pink), and personality (purple) categories of biopsychosocial loadings reflect higher magnitude on these measures. Positive value in the physical health (olive) category represents higher value and positive values in the cognition (orange) category indicate either higher accuracies or slower reaction times (RT). Error bars indicate bootstrapped-estimated confidence intervals (i.e., standard deviation) and measures in bold indicate statistical significance. **(B)** Unthresholded edge-wise beta coefficients obtained from generalized linear models (GLM) between participants' LC3 canonical scores (i.e., averaged sleep and biopsychosocial canonical scores) and their RSFC data; **(C)** FDR-corrected network-wise beta coefficients computed with GLMs within and between 17 large-scale brain networks [46] and subcortical regions [47]. **(D)** Distribution of the integration/segregation ratio in each of the 7 large-scale brain networks and subcortical regions associated with LC3 (left). The dashed line indicates the median of all parcels, and the bold black lines represent the median for each network. The integration/segregation ratio values for the 400 Schaeffer parcellation [48] and 7 subcortical regions are projected on cortical and subcortical surfaces (right). See S1 Data for underlying data.

In brief, most profiles (LCs 1, 4, 5) showed significant associations between sleep-biopsychosocial composite scores and education level, where lower education level was associated with a higher composite score in LCs 1, 4, 5 (all $q < 0.05$). Similarly, lower household income correlated with a higher composite score in LCs 1 and 2 (all $q < 0.05$). Race and ethnicity groups revealed differences in composite sleep and biopsychosocial scores for LCs 1, 3–5 (all $q < 0.05$). Finally, while the presence of a family history of psychopathology was associated with higher biopsychosocial scores in LCs 1 and 2, we only found biological sex differences in LC5, with higher sleep and biopsychosocial composite scores in female participants ($q < 0.05$).

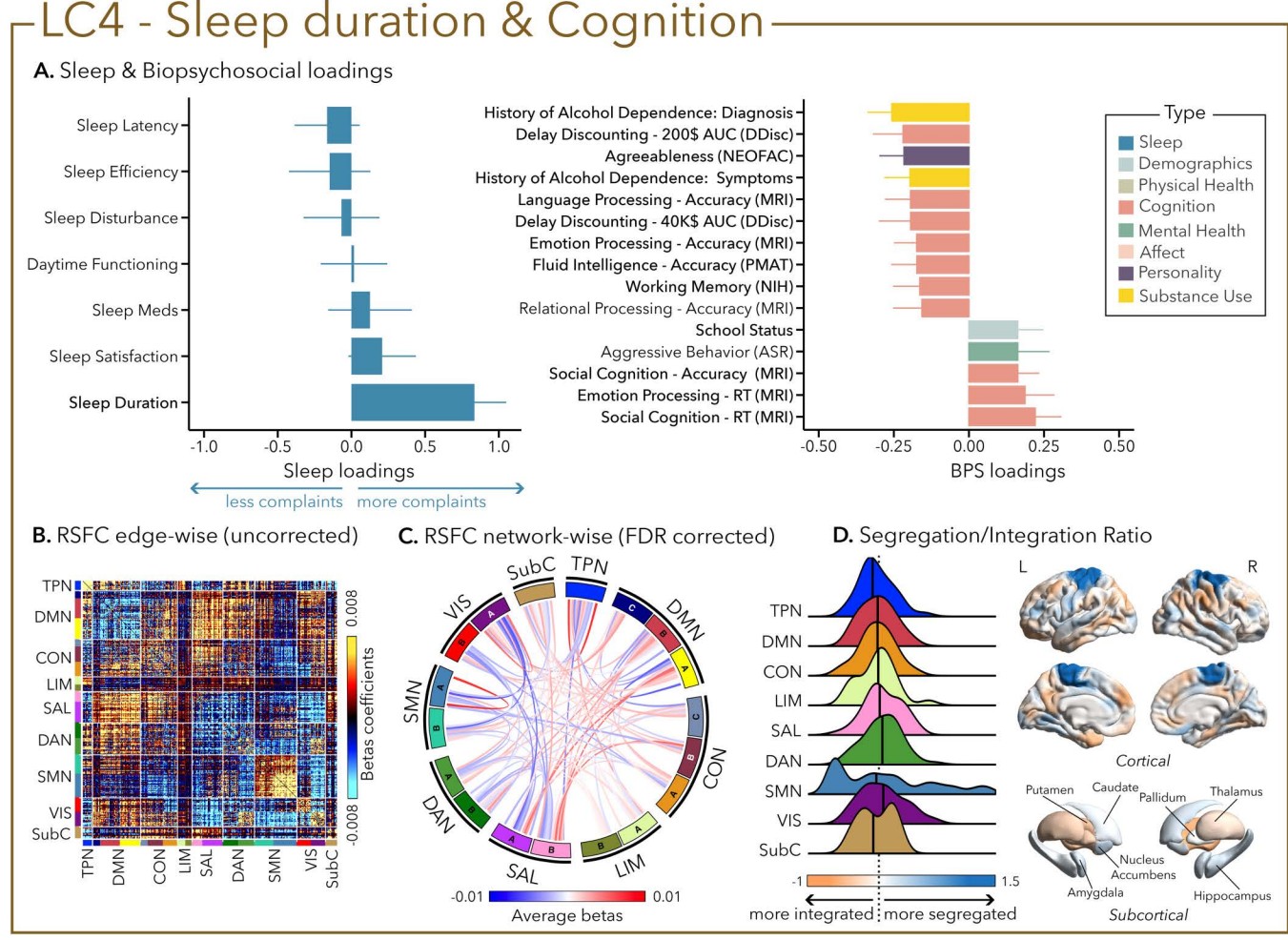

**Fig 5. The fourth latent component (LC4) reflects the association between sleep duration and cognition. (A)** Sleep loadings (left) and top 15 strongest biopsychosocial (BPS) loadings (right) for LC4. Greater loadings on LC4 were associated with shorter sleep duration and measures of poor cognitive performance. Positive values on sleep loadings (blue) indicate worse outcomes while positive values on the mental health (green), substance use (yellow), demographics (light blue), and personality (purple) categories of biopsychosocial loadings reflect higher magnitude on the measures. Positive values in the cognition (orange) category indicate either higher accuracies or slower reaction times (RT). Error bars indicate bootstrapped-estimated confidence intervals (i.e., standard deviation) and measures in bold indicate statistical significance. **(B)** Unthresholded edge-wise beta coefficients obtained from generalized linear models (GLM) between participants' LC4 canonical scores (i.e., averaged sleep and biopsychosocial canonical scores) and their RSFC data; **(C)** FDR-corrected network-wise beta coefficients computed with GLMs within and between 17 large-scale brain networks [46] and subcortical regions [47]. **(D)** Distribution of the integration/segregation ratio in each of the 7 large-scale brain networks and subcortical regions associated with LC4 (left). The dashed line indicates the median of all parcels, and the bold black lines represent the median for each network. The integration/segregation ratio values for the 400 Schaeffer parcellation [48] and 7 subcortical regions are projected on cortical and subcortical surfaces (right). See S1 Data for underlying data.

## Control analyses

We summarize several analyses that demonstrate the robustness of our findings (see Supplemental Results C in S1 Text). First, LC1 and LC2 successfully generalized in our cross-correlation scheme (mean across 5-folds: $r=0.49$, $p=0.001$; $r=0.19$, $p=0.039$ respectively), but not LCs 3–5 (see Table C in S1 Text), suggesting that LCs 3–5 might not be as robust and generalizable, possibly due to these LCs being driven by a single sleep dimension. Second, we re-computed the CCA analysis after: (i) applying quantile normalization on sleep and biopsychosocial measures; (ii) excluding participants

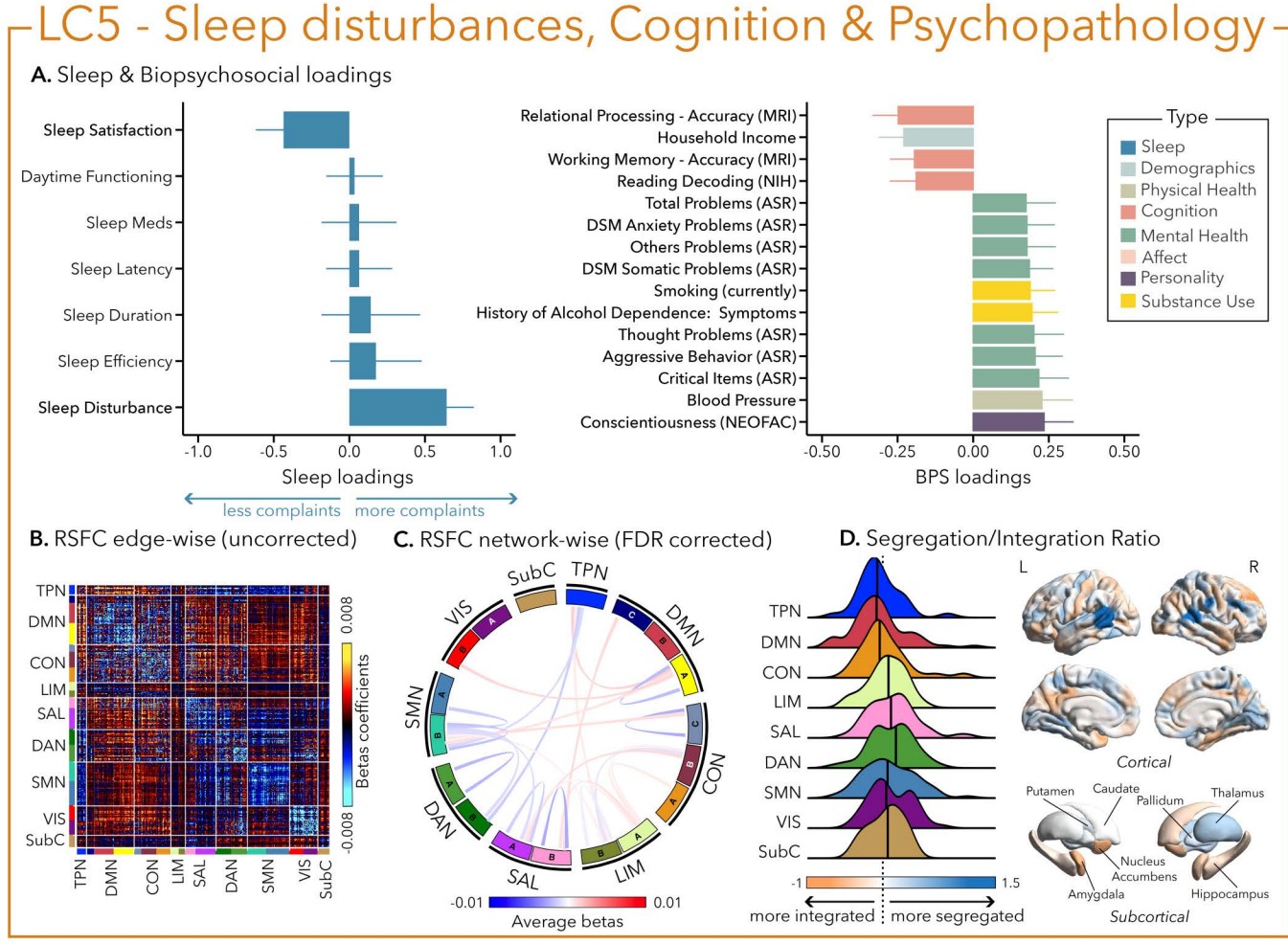

**Fig 6. The fifth latent component (LC5) reflects the association between sleep disturbance, cognition, and psychopathology. (A)** Sleep loadings (left) and top 15 strongest biopsychosocial (BPS) loadings (right) for LC5. Greater loadings on LC5 were associated with the presence of sleep disturbances, higher measures of psychopathology and lower cognitive performance. Positive values on sleep loadings (blue) indicate worse outcomes while positive values on the mental health (green), substance use (yellow), and personality (purple) categories of biopsychosocial loadings reflect higher magnitude on these measures. Positive values in the cognition (orange) category indicate either higher accuracies or slower reaction times (RT), while positive values in the demographics (light blue) and physical health (olive) categories represent higher values. Error bars indicate bootstrapped-estimated confidence intervals (i.e., standard deviation) and measures in bold indicate statistical significance. **(B)** Unthresholded edge-wise beta coefficients obtained from generalized linear models (GLM) between participants' LC5 canonical scores (i.e., averaged sleep and biopsychosocial canonical scores) and their RSFC data; **(C)** FDR-corrected network-wise beta coefficients computed with GLMs within and between 17 large-scale brain networks [46] and subcortical regions [47]. **(D)** Distribution of the integration/segregation ratio in each of the 7 large-scale brain networks and subcortical regions associated with LC5 (left). The dashed line indicates the median of all parcels, and the bold black lines represent the median for each network. The integration/segregation ratio values for the 400 Schaeffer parcellation [48] and 7 subcortical regions are projected on cortical and subcortical surfaces (right). See S1 Data for underlying data.

that had tested positive for any substance on the day of the MRI; (iii) excluding physical health measures (i.e., body mass index, hematocrit, and blood pressure), (iv) excluding sociodemographic variables (i.e., employment status, household income, school status, and relationship status) from the biopsychosocial matrix; (v) after using PCA to reduce the dimensionality of the biopsychosocial variables; and (vi) after considering only female (or male) participants in the CCA. The CCA loadings remained mostly unchanged (Table E in S1 Text). We also assessed the robustness of our imaging results

in several ways. First, we re-computed the GLM analysis using RSFC data that underwent CompCor [49] instead of global signal regression (GSR). The RSFC patterns were altered, although the patterns shared generally high correlations with the main analysis for most of the LCs ($r = 0.75$, $r = 0.76$, $r = 0.78$, $r = 0.51$, and $r = 0.77$ for LCs 1–5, respectively; Fig C in S1 Text). Next, excluding subjects that likely fell asleep in the scanner did not impact our findings ($r = 0.90$, $r = 0.87$, $r = 0.95$, $r = 0.95$, and $r = 0.95$ for LCs 1–5 respectively; Fig C in S1 Text); however, we found that these participants had higher sleep and biopsychosocial composite scores on LC4 compared to participants that likely stayed awake during the scan (Fig D in S1 Text). Finally, we re-computed the GLM analyses by using sleep and biopsychosocial canonical scores instead of averaged scores. We found moderate to high correlations with the main GLM analysis ($r = 0.69$, $r = 0.62$, $r = 0.63$, $r = 0.46$, and $r = 0.67$ for LCs 1–5, respectively; Fig C in S1 Text).

## Discussion

Leveraging a multidimensional data-driven approach in a large cohort of healthy young adults, we uncovered five distinct sleep profiles linked to biopsychosocial factors encompassing health, cognition, and lifestyle. We found that the first profile explained most of the covariance and reflected general psychopathology (or *p factor*) associated with general poor sleep (LC1). The second profile also reflected general psychopathology but in the absence of sleep complaints, which we defined as sleep resilience (LC2). Meanwhile, the three other profiles were driven by a specific dimension of sleep, such as the use of sleep aids (LC3), sleep duration (LC4), or sleep disturbances (LC5), which were associated with distinct patterns of health, cognition, and lifestyle factors. Furthermore, identified sleep-biopsychosocial profiles displayed unique patterns of brain network organization. Our findings emphasize the crucial interplay between biopsychosocial outcomes and sleep, and the necessity to integrate sleep history to contextualize research findings and to inform clinical intake assessments [50].

The dominance of psychopathology markers in most of the profiles is not surprising as the RDoC framework proposed arousal and regulatory systems (i.e., circadian rhythms and sleep/wakefulness) as one of the five key domains of human functioning likely to affect mental health [51], which is consistent with a large literature reporting significant disruption of sleep across multiple psychiatric disorders [8,52]. Although individuals with a neuropsychiatric diagnosis (e.g., schizophrenia or major depressive disorder) were not included in the HCP dataset [38], the presence of the *p* factor, defined as an individual's susceptibility to develop any common form of psychopathology, exists on a continuum of severity and chronicity within the general population [53].

LC1 overwhelmingly explained 88% of the covariance between sleep and biopsychosocial scores in a sample of healthy adults, highlighting the reciprocal relation between sleep and psychopathology and how sleep may be of utmost importance for both the prevention and treatment of mental disorders. LC1 reflected general psychopathology associated with overall poor sleep akin to insomnia complaints (i.e., difficulties falling asleep, maintaining sleep, sleep dissatisfaction, and insufficient sleep). While we could not assess the presence of insomnia disorder based on clinical diagnostic criteria [54,55] or the chronicity of sleep complaints (i.e., PSQI captures sleep complaints only for the previous month) [39], there is a body of evidence of a reciprocal relationship between reports of poor sleep or insomnia complaints and psychopathology [56]. Overall, poor sleep is not only a risk factor but also a co-morbid condition and transdiagnostic symptom for many mental disorders [57]. When sleep is disrupted, it also contributes to the dysregulation of multiple neurobiological mechanisms related to emotional regulation and psychopathology [58]. The strong co-morbidity between poor sleep and psychopathology is evident from a young age, as suggested by a recent study that used a similar data-driven approach (i.e., CCA) to assess associations between parent-reported sleep disturbances and a broad set of psychological, demographics, and cognitive variables in a large sample of 9- to 10-year-old children [59]. The study reported a very similar latent component linking general poor sleep to general psychopathology, and that covariance pattern was replicated after a 2-year follow-up, suggesting the robustness of this association over time.

Although LC1 captured a large amount of covariance between sleep and biopsychosocial measures, LCs 2–5 showed covariance patterns that were characterized by specific sleep dimensions, which likely described associations seen in a fraction of the participants; alternatively, these associations might also be present in all or most participants, but with less prominence. Symptoms of psychopathology mirrored each other across LC1 and LC2, but the paradoxical contrast in sleep loadings suggests that some individuals might have more resilient sleep (LC2), whereby they might be able to maintain healthy sleep patterns in the face of psychopathology. However, the cause of such resilience is unclear. Up to 80% of individuals experiencing an acute phase of mental disorder (e.g., depressive and/or anxiety episode) report sleep issues [8,60,61], leaving a minority of individuals who do not report abnormal sleep during such episodes. The identification of LC2 supports this and suggests there might be biological or environmental protective factors in some individuals who would otherwise be considered at risk for sleep issues. However, our understanding of such protective factors is limited [62–64]. Another possible interpretation is that LC2 might reflect individuals who may lack insights into their sleep difficulties, particularly relative to other concerns that are more burdensome to them (i.e., difficulties functioning during daytime). Indeed, some individuals often fail to recognize the full impact of their sleep disturbances and attribute their daytime symptoms to external factors or normalize their tiredness [65]. Nonetheless, whether this profile of sleep resilience or sleep misperception is a stable latent component, a cross-sectional observation of fluctuating symptoms that may develop into psychopathology-related sleep complaints or underlie objective sleep alterations, needs to be further tested.

Interestingly, distinctions between LC1 and LC2 were also present in the neural signatures of RSFC, which may assist in the neurobiological interpretation of the profiles. Visually inspecting LC1 and LC2 suggested an underlying increase in subcortical-cortical connectivity when sleep disturbances are associated with psychopathology. This is in alignment with the known neurophysiology of the ascending arousal system and possibly implies the existence of some level of hyperarousal in these pathways that may contribute to disturbances in sleep [66]. The stronger RSFC between subcortical, somatomotor, and dorsal attention networks (DAN) in LC1, and the preserved thalamocortical and control network coupling in LC2, nevertheless suggest that these network patterns may reflect vulnerability and resilience pathways to sleep disturbances in psychopathology. However, this speculation requires further targeted research to be confirmed. The reduced connectivity between the TPN and both the dorsal attention and somatomotor networks in LC1 may reflect a breakdown in the typical antagonism between internally and externally oriented brain systems [67], potentially facilitating maladaptive self-referential processing. This pattern aligns with prior work linking increased default mode network (DMN) connectivity to rumination in depression [68,69] and extends these findings by suggesting that such connectivity profiles may underlie transdiagnostic vulnerability to poor sleep quality. The absence of this pattern in LC2—despite similar levels of psychopathology—raises the possibility that disrupted TPN-DAN balance is not merely a consequence of mental health symptoms but may reflect a mechanism contributing specifically to sleep disturbance.

Within the profiles driven by a specific sleep dimension, LC5 also reflected some dimensions of psychopathology (i.e., anxiety, thought problems) that were only associated with the presence of sleep disturbances. The sleep disturbance sub-component of the PSQI is broad and encompasses complaints of sleep-related breathing problems as well as multiple awakenings that could be due to nycturia, pain, nightmares, or difficulties maintaining optimal body temperature [39]. Altogether, the sleep disturbances dimension is thought to represent sleep fragmentation [70]. This covariance pattern is in line with a recent study conducted in a large community-based cohort (i.e., UK Biobank) that found that lifetime diagnoses of psychopathology and psychiatric polygenic risk scores were more strongly associated with accelerometer-derived measures of sleep quality (i.e., fragmentation) than sleep duration per se [71]. Interestingly, biological sex differences emerged only in LC5, with female participants showing higher sleep and biopsychosocial scores than male participants. Such differences often arise at puberty, with female participants reporting more sleep fragmentation (i.e., awakenings, time spent awake in the middle of the night) [72] and higher rates of depression and anxiety across the life span (i.e., reproductive events) [73] compared to male participants. While there was no sex difference in the other LCs, the sex specificity of LC5 highlights the meaningful interplay between biological sex and individual differences in sleep disturbances and mental health.

We found that sleep duration (driving LC4) was not associated with measures of psychopathology but rather with cognitive performance (e.g., reduced accuracy in working memory, emotional processing, and language processing). Whether studied via experimental acute sleep deprivation, chronic sleep restriction, or in clinical populations (e.g., insomnia with objective short sleep duration), the consequences of lack of sleep on daytime functioning and health are well known and substantial [11,12,18,74–76]. Sleep duration affects, in varying effect sizes, both accuracy and reaction time in most cognitive tasks [11,12,75]. Interestingly, the strong RSFC patterns associated with LC4 showed a global increase in connectivity, alongside localized segregation of part of the somatomotor network. Similar patterns have been reported in neuroimaging studies of experimental acute total sleep deprivation [77,78] and were recently found to be associated with sleep duration in adolescents [16]. These RSFC features are thought to reflect homeostatic mechanisms that regulate brain function and suggest that LC4 may index underlying sleep debt in the general population.

Finally, beyond sleep measures and sleep-related daytime functioning, the PSQI also evaluates the use of medication to help sleeping [39], whether prescribed or over-the-counter (e.g., gamma-aminobutyric acid $GABA_A$ receptor modulators, selective melatonin receptor agonists, selective histamine receptor antagonists, cannabinoid products, valerian) [79]. We found that LC3 was driven by the use of sleep aids and was mostly associated with reports of satisfaction in social relationships. This profile specifically highlights a subgroup of young adults who appear to manage sleep difficulties with pharmacological solutions. As such, the associated biopsychosocial factors, in particular high sociability, could result from the effect of the drug itself on social behavior and positive mood (e.g., via potentiation of GABA transmission) [80,81] or as a consequence of the drugs on sleep complaints [82], which may support better well-being, and consequently translate to greater satisfaction in social relationships and support systems [82,83]. However, as the PSQI only covers the past month, we lacked data on the type and duration of use, limiting insights into long-term cognitive effects [84,85] or the development of substance abuse that have been documented. LC3, defined by sleep-aid use and relative absence of daytime complaints, showed increased RSFC within the visual and default mode networks, and greater segregation in visual and limbic systems. This pattern, along with impaired visual memory and emotion recognition performance, may reflect sedation-related reductions in network integration [86] that disrupt perceptual and affective-cognitive processes, despite subjectively intact attention and daytime functioning.

Interestingly, alterations to the segregation/integration ratio of the somatomotor and visual cortex were common in most profiles. Highly interconnected to the whole brain, the somatomotor network is crucial for processing external stimuli and producing motor responses, but is also functionally involved in bodily self-consciousness and interoception. Altered dysconnectivity patterns of the somatomotor network have been linked to variation in several domains, including general psychopathology [87,88], cognitive dysfunction related to sleep deprivation [77], as well as the total PSQI score [13,89]. Overall, these findings suggest that alterations to RSFC in the somatomotor network are also involved in the relationships between sleep and biopsychosocial factors and highlight the importance of better understanding the role of this brain network in overall health and functioning.

These profiles contribute to a deeper understanding of the current debate that opposes sleep quality and sleep duration [7,90]. In line with previous studies [11,12,91], we found that cognitive functioning was more related to sleep duration than subjective sleep quality; in addition, we found that sleep disturbances, alone (LC5) or in combination with other sleep complaints (LC1) were strongly associated with self-reported psychopathology. Moreover, it is also important to note that complaints of poor sleep quality and/or short sleep duration have been both associated with increased risk of physical health outcomes and all-cause mortality [6,7]. While LC1 and LC2 presented sleep dimensions as being inextricably linked, LC3, LC4, and LC5 revealed distinct facets of sleep, suggesting that while sleep dimensions are related, they can also be separable domains with specific links to biopsychosocial factors. This is likely reflected in the finding that only LC1 and LC2 were replicable in cross-validation analyses, which may be due to LC3, LC4, and LC5 being driven by a single sleep dimension and thus contributing only marginally to the variance.

While unidimensional association studies are informative, our findings reinforce the notion that sleep health is multidimensional and distinct measures of sleep quantity or quality should be considered together when investigating their influence on biopsychosocial aspects of health, cognition, and lifestyle. The use of multivariate approaches provides insights into the multidimensional nature of sleep and/or biopsychosocial outcomes [15,21,30–32,59,92]. However, it is important to note that our findings, driven by CCA and RSFC correlations, do not inform on the directionality or causality of these effects. Future work is needed to extend these findings and further explore the multidimensional nature of sleep health, for instance, taking into consideration the U-shaped relationship between sleep duration and biopsychosocial measures [63,93,94]. Given the design of the PSQI, only short sleep duration (<5–6 h) was considered as a sleep difficulty, neglecting the potential consequences of long sleep duration (>9 h). Long sleep duration is commonly observed in hypersomnia disorders and psychopathology (e.g., schizophrenia, depression) [6,95], as well as being associated with increased risk of cardiovascular heart disease and mortality [7,96,97], depression and cognitive decline [6,22,63,94]. This U-shape relationship, whereby both short and long sleep durations are associated with negative impact on health and cognition as well as increasing markers of cerebrovascular burden (e.g., white matter hyper-intensities) [63,93], may provide a window to identify mechanisms that underlie the interplay between sleep and biopsychosocial factors.

Other considerations moving forward include sleep regularity and sleep timing, which are not part of the computation of the sub-components of the PSQI [39]; hence, their association with biopsychosocial outcomes was not investigated in this study. Furthermore, the PSQI is often interpreted with regard to its total score (combining all sub-components), which provides a binary vision of sleep quality (i.e., either good or bad sleep) [39]. In this study, we did not want to be limited by the PSQI global score but rather aimed to untangle the different dimensions (or sub-components) of sleep and their relationship to biopsychosocial and neurobiological measures.

Furthermore, whether or not participants fell asleep in the scanner did not impact our RSFC findings; however, the choice of preprocessing may have had repercussions on our results. The current state of neuroimaging suffers from a lack of consistency and agreement on preprocessing techniques, and these have been shown to alter relationships between RSFC and behavior [98]. While we acknowledge this limitation, a comprehensive investigation into the impact of preprocessing is beyond the scope of this paper. In addition, the time-of-day of the 2 sessions of fMRI acquisition, which has been shown to also affect relationships between RSFC and behavior [99], could have also impacted our RSFC findings. Finally, we chose to include participants who were likely to have consumed psychoactive drugs on the day of the fMRI acquisition, as we favored a more naturalistic analysis; however, we showed that excluding these participants did not have any impact on our findings. Moreover, the RSFC patterns we found had been previously described in the sleep literature, suggesting that they are likely robust to these effects.

A final important distinction to be addressed is that sleep and biopsychosocial outcomes were mostly self-reported through questionnaires. Both objectively recorded and subjectively perceived estimations provide different yet meaningful information that tends to positively correlate [100]. However, it has been shown that when compared to objective estimates (i.e., polysomnography and/or actigraphy recordings), individuals with sleep complaints (i.e., chronic insomnia, obstructive sleep apnea) tend to subjectively misperceive their sleep (i.e., duration, sleep latency) [27,28,101,102]. The degree of discrepancy between objective and subjective measures (i.e., sleep state misperception) has been correlated with worse sleep quality [103,104] as well as compromised reports of daytime functioning [26]. While objective measurements might have exposed divergent associations between sleep and biopsychosocial factors, the profiles reported here arguably support greater clinical validity, as subjective complaints are often what drives an individual to seek out healthcare. Our study emphasizes that considering individuals' sleep experience can support clinicians to make more accurate initial assessments and navigate the course of treatment and interventions. It also paves the way for future research to examine the LCs reported here using more objective measures of sleep.

The awareness and interest surrounding sleep as a crucial pillar of health are growing rapidly [105]. However, the role of sleep in general health is complex, multifaceted, and largely unknown. The multidimensional approach applied in this large sample of healthy young adults is a first step that we argue should be implemented in future research incorporating sleep. We highlight the observation of five distinct sleep patterns associated with specific combinations of biological, psychological, and socio-environmental factors, related to distinct brain connectivity patterns. Nonetheless, our findings would benefit from including a more diverse sample of participants with specific clinical concerns (whether in terms of sleep and/or biopsychopathology). These findings support the notion that sleep is emerging as a distinguishable factor that can assist in disentangling the complex heterogeneity of human health. As the capacity for large-scale human research continues to grow, integrating sleep dimensions at such a scale is not only feasible in terms of evaluation but presents a unique opportunity for translational application. Sleep is a modifiable lifestyle factor and can be investigated in model organisms as well as in humans, and as such is well-positioned to identify potential converging mechanisms and intervention pathways or tools. The current study emphasizes that by using a multidimensional approach to identify distinct sleep-biopsychosocial profiles, we can begin to untangle the interplay between individuals' variability in sleep, health, cognition, lifestyle, and behavior—equipping research and clinical settings to better support individuals' well-being. Future investigations into how the multifaceted relationships between sleep and biopsychosocial factors differ or change according to age, sex, and other demographics would likely benefit from data-driven approaches.

## Materials and methods

### Participants

Data for this study were obtained from the S1200 release of the publicly available HCP dataset [38]. The WU-Minn HCP Consortium obtained full informed written consent from all participants. Research procedures and ethical guidelines were followed per Washington University institutional review board approval and experiments were conducted following the ethical principles outlined in the Declaration of Helsinki (see [38]). Our use of the HCP dataset for this study was carried out with local institutional review board approval at the National University of Singapore (N-17-056). The HCP dataset comprises multimodal MRI data, including structural MRI, diffusion MRI, resting-state, and task functional MRI (fMRI) data, as well as a broad range of behavioral measures collected in young healthy subjects (aged 22–36). Details about imaging acquisition parameters and data collection [38], as well as the list of available behavioral and demographics measures (HCP S1200 Data Dictionary) [106] can be found elsewhere. Of note, the HCP dataset comprises a large number of related individuals (i.e., siblings and twins). Of the 1,206 total subjects available from the HCP S1200 release, we excluded 403 participants with missing/incomplete data on one or more measures of interest, and 33 participants with visual impairment that might have impacted their task performance in the scanner. Our final sample comprised 770 participants (53.76% female, 28.86±3.61 years old). We decided to keep participants ($N = 94$) who tested positive for any substance (including alcohol, marijuana, and other drugs) on the day of the MRI, as substance use has intricate links to sleep, and we did not want to exclude the possibility of finding potential substance use-related sleep profiles. However, we also re-computed our analyses after excluding these individuals ($N = 676$) and found very similar results (see Table E in S1 Text). Out of these 770 participants, 723 passed MRI quality control and were included in the *post hoc* RSFC analyses.

### Sleep assessment

Participants were administered the PSQI [39] to assess different aspects of their sleep over the past month. To define sleep in our study, we used the 7 sub-components of the PSQI which characterize different sleep dimensions, namely (i) sleep satisfaction, (ii) sleep latency, (iii) sleep duration, (iv) sleep efficiency, (v) sleep disturbance, (vi) sleep-aid medication, and (vii) daytime functioning. Sub-components are calculated through 4 questions on the timing of sleep habits and 6 Likert-scale questions from 0 to 3, 0 being best and 3 being worst.

## Biopsychosocial assessment

One hundred eighteen biopsychosocial measures were selected from the HCP dataset (see complete list in Table A in S1 Text). These measures included self-reported assessments of current and past mental health and substance use, questionnaires on personality, affect, lifestyle, and demographics, cognitive tasks tapping on different processes such as working memory or social cognition performed either inside or outside the MRI, and physical assessments (e.g., blood pressure). These measures did not undergo any dimensionality reduction or clustering by biopsychosocial domain in order to preserve granularity in the way they would be associated with sleep dimensions. Biopsychosocial measures with large amounts of missing data were excluded, as well as similar measures that were likely to be redundant. Biopsychosocial measures were categorized by behavioral domain (e.g., cognition, physical health) based on the way they had been described in the HCP dataset [38,106].

## Canonical correlation analysis

CCA [107,108], a multivariate data-driven approach, was applied to the sleep and biopsychosocial measures, after regressing out the effects of age, sex, and education from both the sleep and biopsychosocial variables. CCA derives latent components (LCs, i.e., canonical variates), which are optimal linear combinations of the original data, by maximizing *correlation* between two data matrices (i.e., sleep and biopsychosocial measures).

We applied the *canoncorr* function from Matlab 2018b to our dataset and the CCA analysis was computed as follows. Sleep and biopsychosocial measures are stored in matrices $X$ (770 × 7) and $Y$ (770 × 118). First, $X$ and $Y$ each undergo orthogonal decomposition such that:

$$X = Q1 \times R1$$

$$Y = Q2 \times R2$$

where $Q1$ and $Q2$ are orthogonal matrices, and $R1$ and $R2$ are upper unitary matrices. Orthogonal matrices are then multiplied to obtain a correlation matrix:

$$Q = Q1^T \times Q2$$

Onto which singular value decomposition (SVD) is applied:

$$Q = A \times S \times B^T$$

This results in two singular vector matrices, $A$ and $B$, and a diagonal matrix containing the singular values, $S$. The singular vector matrices of each LC form the sleep weights (7 × 7), and biopsychosocial weights (118 × 7). When $A$ and $B$ are linearly projected onto respective sleep and biopsychosocial scores, $X$ and $Y,$ it yields maximally correlated canonical variates:

$$U = X \times A$$

$$V = Y \times B$$

The rank of the correlation matrix determines the number of derived LCs (i.e., in this case, the number of sleep measures, hence 7 LCs). Each sleep-biopsychosocial LC is characterized by a pattern of sleep weights and a corresponding

pattern of biopsychosocial weights (i.e., canonical coefficients). Linear projection of sleep (or biopsychosocial) data onto sleep (or biopsychosocial) weights yielded participant-specific composite scores for sleep (or biopsychosocial) measures (i.e., canonical scores). The contribution of original sleep and biopsychosocial loadings to each LC was determined by computing Pearson's correlations between sleep (or biopsychosocial) data and participant-specific scores for sleep (or biopsychosocial factors) to obtain sleep and biopsychosocial *loadings* (i.e., canonical structure coefficients) [109,110]. Canonical structure coefficients reflect the direct contribution of a predictor (e.g., one sleep dimension) to the predictor criterion (e.g., LC1) independently of other predictors (e.g., LCs 2–7), which can be critical when predictors are highly correlated with each other (i.e., in presence of multicollinearity) [111]. We did not employ dimensionality reduction (e.g., via principal components analysis), as the sample size ($N = 770$) exceeded the number of sleep (7 measures) and bio-psychosocial measures (118 measures) being modeled. Statistical significance of each of the 7 LCs was determined by permutation testing (10,000 permutations) followed by FDR correction. Given the high prevalence of related participants in the HCP dataset, family structure was maintained during permutations (using the PALM package [112,113]), whereby monozygotic twins, dizygotic twins, and nontwin siblings were only permuted within their respective groups. Finally, the loadings' stability was determined using bootstrap resampling to estimate confidence intervals for the loadings, by deriving 1,000 samples with replacement from participants' sleep and biopsychosocial data.

## MRI acquisition and processing

All imaging data were acquired on a customized Siemens 3T Skyra scanner at Washington University (St Louis, MI, USA). Four runs of resting-state fMRI were collected over two sessions across two separate days. Each run included 1,200 frames using a multi-band sequence at 2-mm isotropic spatial resolution with a TR of 0.72 s for 14.4 min. The structural images were acquired at 0.7-mm isotropic resolution. Further details of the data collection and HCP preprocessing are available elsewhere [38,114,115]. Notably, cortical and subcortical data underwent ICA-FIX [116,117] and were saved in the CIFTI gray ordinate format. The surface (fs_LR) data were aligned with MSM-All [118]. As ICA-FIX does not fully eliminate global motion-related and respiratory-related artifacts [119,120], additional censoring and nuisance regression were performed [98,121]. In particular, volumes with framewise displacement (FD) > 0.2 mm, and root-mean-square of voxel-wise differentiated signal (DVARS) > 75 were marked as outliers and censored, along with one frame before and two frames after the outlier volume [122,123]. Any uncensored segment of data that lasted fewer than five contiguous volumes was also excluded from analysis, as well as runs with >50% censored frames. Additionally, the global signal obtained by averaging signal across all cortical vertices and its temporal derivatives (ignoring censored frames) was also regressed out from the data because previous studies have suggested that global signal regression strengthens the association between RSFC and behavioral traits [98]. As there is ongoing debate on the use of GSR as a means of fMRI preprocessing [98,124–126], additional reliability analysis was performed on data preprocessed using a component-based noise correction method (CompCor) [49] instead of GSR.

RSFC was computed among 400 cortical parcels [48] and 19 subcortical regions [47] using Pearson's correlation (excluding the censored volumes). The subcortical regions were in subject-specific volumetric space as defined by FreeSurfer [47], and comprised the left and right cerebellum, thalamus, caudate, putamen, pallidum, hippocampus, accumbens, amygdala, ventral diencephalon, and brainstem. For each participant, RSFC was computed for each run, Fisher z-transformed, and then averaged across runs and sessions, yielding a final 419×419 RSFC matrix for each participant.

## RSFC analyses

To investigate the neurobiological substrates of the sleep-biopsychosocial profiles derived in the CCA, we computed generalized linear models (GLM) between participants' canonical scores (i.e., averaged sleep and biopsychosocial scores) and their RSFC data. Age, sex, and level of education were first regressed out from the RSFC data.

To obtain an analysis at the large-scale network level and limit the number of multiple comparisons, we computed a network-wise GLM, whereby the whole-brain RSFC data were averaged within and between the 17 large-scale brain networks [48] and subcortical regions [47], resulting in 18 × 18 RSFC matrices. Next, we applied a GLM for each network edge (i.e., average connectivity between two brain networks), with participants' component-specific canonical scores as the predictor and RSFC edge as the response. Each GLM yielded a beta coefficient and associated $T$ statistic, as well as an $F$ statistic and associated $p$ value obtained from a hypothesis test that all coefficient estimates were equal to zero. Statistical significance for each RSFC network edge was determined by applying FDR correction ($q < 0.05$) on all $p$ values (along with other *post hoc* analyses). For a more granular view, we also computed a GLM for each RSFC edge (i.e., connectivity between two brain regions) using whole-brain RSFC between all 419 brain regions. For a complete view of the component-specific RSFC signatures, we plotted both the uncorrected region-wise GLM beta coefficients (e.g., Fig 2B) and FDR-corrected network-wise GLM beta coefficients (e.g., Fig 2C).

Measures of integration and segregation were computed on the GLM beta coefficient connectivity matrix associated with each LC using functions from the Brain Connectivity Toolbox [127]. Firstly, the input-weighted connection matrix was normalized. Next, each 419 cortical parcel was assigned to one of the 7 large-scale brain networks and subcortical regions [46]. Within-network connectivity was estimated by calculating the module-degree Z score (within-module strength) for each region. The extent to which a parcel connects across all networks was quantified using the participation coefficient, (between-module strength). For each cortical parcel, the ratio of normalized within:between module strength values was calculated and interpreted as a measure of the balance of integration and segregation of functional brain connectivity [128]. Nodes with high within but low between-module strength are likely to facilitate network segregation, while nodes with higher between-module strength (i.e., connector hubs) are likely to facilitate global integration [127].

## Control analyses

We ran several control analyses to evaluate the robustness of our findings. First, we applied 5-fold cross-validation (accounting for family structure) to assess the generalizability of our sleep-biopsychosocial profiles by training a CCA model on 80% of the data and testing it on the remaining 20% of the data. For each fold, we projected the sleep and biopsychosocial canonical coefficients of the training data on the sleep and biopsychosocial data of the test data, to obtain sleep and biopsychosocial scores, and computed Pearson's correlations between these scores. Second, we evaluated the impact of the covariates on our profiles as well as the impact of other potential confounds, including race, ethnicity, and familial psychiatric history. Third, we re-computed the CCA after excluding participants who had tested positive for any substance use on the day of the MRI. Fourth, we re-computed the CCA after excluding physical health (i.e., body mass index, hematocrit, and blood pressure) and sociodemographic (i.e., employment status, household income, in-school, and relationship status) variables from the biopsychosocial matrix. Fifth, to mitigate scale magnitude discrepancies between different measures, we re-computed the CCA after applying quantile normalization on sleep and biopsychosocial measures. Sixth, to test the stability of our LCs, we re-computed the CCA after reducing the dimensionality of the biopsychosocial variables and using the principal components that explained 90% of the variance among the 118 biopsychosocial variables. Next, we re-computed the CCA within female or male participants only. We also assessed the robustness of our imaging results in several ways. As GSR is a controversial preprocessing step [98,125,126], we re-computed the GLM analysis using RSFC data that underwent CompCor [49] instead of GSR. Some subjects were noticed to have likely fallen asleep during scanning (list not publicly available [129]). As a first step, we re-computed the GLM after excluding these subjects ($N = 100$); next, we sought to determine whether these participants scored high on any of the profiles, by comparing their sleep/biopsychosocial composite scores with awake participants using t-tests. We re-computed the GLM analyses by using sleep and biopsychosocial canonical scores instead of averaged scores. Finally, integration and segregation measures were also computed on the average RSFC matrix of the whole sample. FDR correction ($q < 0.05$) was applied to all *post hoc* tests.

## Supporting information

**S1 Data.** Sheet 1. Table of content. Sheet 2. Data for Table 1. Sheet 3. Data for Figs 2A, 3A, 4A, 5A, 6A, S1A (sleep). Sheet 4. Data for Figs 2B, 3B, 4B, 5B, 6B, S1B (biopsychosocial). Sheet 5. Data for Figs 1, 2, 3, 4, 5, 6, S1 (subject). Sheet 6. Data for Fig S4. Sheet 7. Data for Fig S7.
(XLSX)

**S1 Text.** **Table A**. Sleep and nonsleep-biopsychosocial measures used in the CCA (indicated in black), or in post hoc analyses (indicated in blue), or in both (indicated in green). **Table B.** CCA loadings and z-scores for sleep and biopsychosocial measures for LCs 1–5. Loadings with significant bootstrapped z-scores that survived FDR correction ($q < 0.05$) are indicated in bold. **Table C.** Cross-validated canonical correlation analysis. Significant $p$-values are indicated in bold. **Table D.** Post hoc associations between sleep (or biopsychosocial; BPS) composite scores and sociodemographic measures, as well as physical and mental health measures. Associations using continuous measures (e.g., age, education years) were tested with Pearson correlations, while categorical measures (e.g., sex, race) were assessed with $t$-tests or analyses of variance. Associations indicated in bold were found to be statistically significant after FDR correction ($q < 0.05$). Tests with <5 subjects in a category were not computed because of too little variance. **Table E.** Correlations between sleep (or biopsychosocial) loadings in control analyses with those from the original analysis. **Fig A.** Latent Components LC6 and LC7. **(A)** Sleep loadings (left) and top 15 strongest biopsychosocial (BPS) loadings (right) for LC6. **(B)** Sleep loadings (left) and top 15 strongest biopsychosocial (BPS) loadings (right) for LC7. Positive values on sleep (blue) loadings indicate worse outcomes while positive values on biopsychosocial loadings reflect higher magnitude on these measures. Error bars indicate bootstrapped-estimated confidence intervals (i.e., standard deviation) and measures in bold indicate statistical significance. **Fig B.** Post hoc associations between CCA mean composite scores and RSFC, highlighting beta coefficients between subcortical regions and cortical networks. **Fig C.** Control analyses for post hoc associations with RSFC. (Left panel) GLM analysis using RSFC data that underwent CompCor instead of global signal regression. (Middle left panel) GLM analysis after excluding subjects that likely fell asleep in the scanner ($N = 100$). (Middle right panel) GLM analysis between RSFC and sleep composite scores (instead of averaged composite scores). (Right panel) GLM analysis between RSFC and biopsychosocial composite scores (instead of averaged composite scores). Abbreviations: CON, Executive control network; DAN, Dorsal attention network; DMN, Default mode network; LIM, Limbic network; SAL, Salience/Ventral attention network; SMN, Somatosensory-motor network; SubC, Subcortical regions; TPN, Temporoparietal network; VIS, Visual network. **Fig D.** Post hoc $t$-tests assessing differences in sleep (or biopsychosocial) composite scores between participants who likely stayed awake ($N = 623$) in the scanner versus those who likely fell asleep in the scanner ($N = 100$). **Fig E.** Distribution of the segregation and integration ratio on the average RSFC matrix of the whole sample compared to integration and segregation measures per LC. The dashed line indicates the median of all parcels, and the bold black lines represent the median for each network. **Fig F.** Scatterplots showing the distribution of sleep and biopsychosocial composite scores of participants who likely fell asleep in the scanner (in red) compared to those who likely stayed awake (in gray). **Fig G.** Post hoc analyses testing for sex differences in the sleep and behavior composite scores between male (green) and female (red) participants. Male participants had significantly lower sleep ($t = -2.88$, $p = 0.004$) and biopsychosocial ($t = -4.35$, $p < 0.001$) scores for LC5 compared to female participants.
(PDF)

## Acknowledgments

Any opinions, findings, conclusions, or recommendations expressed in this material are those of the authors and do not reflect the views of the Singapore NRF, Singapore NMRC, MOH, or Temasek Foundation. Our research also utilized resources provided by the Center for Functional Neuroimaging Technologies, P41EB015896 and instruments supported by 1S10RR023401, 1S10RR019307, and 1S10RR023043 from the Athinoula A. Martinos Center for Biomedical Imaging

at the Massachusetts General Hospital. The computational work was partially performed using resources of the National Supercomputing Centre, Singapore (http://www.nscc.sg). Data were provided by the Human Connectome Project, WU-Minn Consortium (Principal Investigators: David Van Essen and Kamil Ugurbil; 1U54MH091657) funded by the 16 NIH Institutes and Centers that support the NIH Blueprint for Neuroscience Research; and by the McDonnell Center for Systems Neuroscience at Washington University. Finally, we thank Dr. Joshua Gooley for his helpful comments on the previous versions of the work.

## Author contributions

**Conceptualization:** Aurore A. Perrault, Valeria Kebets, Nicole M. Y. Kuek, B. T. Thomas Yeo.

**Data curation:** Aurore A. Perrault, Valeria Kebets, Nicole M. Y. Kuek, Jingwei Li.

**Formal analysis:** Aurore A. Perrault, Valeria Kebets, Nicole M. Y. Kuek, Nathan E. Cross.

**Funding acquisition:** B. T. Thomas Yeo.

**Methodology:** Valeria Kebets, Nicole M. Y Kuek, Nathan E. Cross.

**Project administration:** B. T. Thomas Yeo.

**Visualization:** Aurore A. Perrault, Valeria Kebets, Nathan E. Cross.

**Writing – original draft:** Aurore A. Perrault, Valeria Kebets, Nicole M. Y. Kuek.

**Writing – review & editing:** Aurore A. Perrault, Valeria Kebets, Nathan E. Cross, Rackeb Tesfaye, Florence B. Pomares, Jingwei Li, Michael W. L. Chee, Thien Thanh Dang-Vu, B. T. Thomas Yeo.

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
