## [Editor Report · Decision Letter 0]

20 Sep 2024

Dear Dr Perrault,

Thank you for submitting your manuscript entitled "A multidimensional investigation of sleep and biopsychosocial profiles with associated neural signatures" for consideration as a Research Article by PLOS Biology.

Your manuscript has now been evaluated by the PLOS Biology editorial staff as well as by an academic editor with relevant expertise and I am writing to let you know that we would like to send your submission out for external peer review.

Once your full submission is complete, your paper will undergo a series of checks in preparation for peer review. After your manuscript has passed the checks it will be sent out for review. To provide the metadata for your submission, please Login to Editorial Manager (https://www.editorialmanager.com/pbiology) within two working days, i.e. by Sep 22 2024 11:59PM.

Kind regards,

Christian

Christian Schnell, PhD

Senior Editor

PLOS Biology

cschnell@plos.org

---

## [Decision Letter · Decision Letter 1]

20 Dec 2024

Dear Dr Perrault,

Thank you for your patience while your manuscript "A multidimensional investigation of sleep and biopsychosocial profiles with associated neural signatures" was peer-reviewed at PLOS Biology. I am sorry that is taken such a long time to get back to you with our decision. The delay was caused first by difficulties in finding suitable reviewers and then reviewers not being able to submit their reports anymore, so we had to recruit new reviewers. In any case, your manuscript has now been evaluated by the PLOS Biology editors, an Academic Editor with relevant expertise, and by several independent reviewers.

In light of the reviews, which you will find at the end of this email, we would like to invite you to revise the work to thoroughly address the reviewers' reports.

As you will see below, the reviewers agree that the study is interesting and provides potentially important insights. Reviewer 1 has one statistical concern and requests a more thorough discussion of the literature. Reviewer 2 asks for a few clarifications, more extensive discussions, and additional analyses of the existing data. Reviewer 3 lists a few points were the evidence is not sufficiently strong and requires further support.

Given the extent of revision needed, we cannot make a decision about publication until we have seen the revised manuscript and your response to the reviewers' comments. Your revised manuscript is likely to be sent for further evaluation by all or a subset of the reviewers.

**IMPORTANT - SUBMITTING YOUR REVISION**

*Re-submission Checklist*

*Published Peer Review*

*PLOS Data Policy*

*Blot and Gel Data Policy*

Kind regards and happy holidays,

Christian

Christian Schnell, PhD

Senior Editor

PLOS Biology

cschnell@plos.org

REVIEWS:

Reviewer #1: In this work, the authors used CCA in the HCP dataset to derive latent components describing the multivariate relations between sleep and cognition/psychopathology factors. They then related these latent components to the brain functional connectivity profiles of the participants. I generally like this paper and do not have many comments. I did want to see some discussion on the following:

1. Even though the sample size is greater than the number of variables (n > p) here, it is not more than 10 times larger (my own simulations have shown n > 12*p works well for CCA). I was wondering if some dimension reduction might be a good further sensitivity analysis to perform.

2. The dominance of the LC1 over the other LCs in the attributed covariance should be brought to the attention of the readers more and discussed a bit (and maybe temper some of the conclusions about the additional LCs).

3. Discussion paragraph 5: A recent preprint relates sleep and biopsychosocial factors to somatomotor disconnect in the ABCD dataset [1]. I was wondering if the SMN being more integrated than segregated in the LC1 in this study is the opposite of the findings from the ABCD in [1]? If that is the case, please discuss this.

[1] Michael, C., Taxali, A., Angstadt, M., McCurry, K. L., Weigard, A., Kardan, O., ... & Sripada, C. (2024). Somatomotor disconnection links sleep duration with socioeconomic context, screen time, cognition, and psychopathology. bioRxiv, 2024-10.

4. Discussion paragraphs 2 and later 6: Some more recent works to potentially discuss if relevant: There is recent work relating sleep disturbances to different psychosocial domains in the ABCD data using CCA [1], and work using both HCP and ABCD datasets on relations between brain functional profiles and sleep duration [2].

[1] McCurry, K. L., Toda-Thorne, K., Taxali, A., Angstadt, M., Hardi, F. A., Heitzeg, M. M., & Sripada, C. (2024). Data-driven, generalizable prediction of adolescent sleep disturbances in the multisite Adolescent Brain Cognitive Development Study. Sleep, 47(6).

[2] Mummaneni, A., Kardan, O., Stier, A. J., Chamberlain, T. A., Chao, A. F., Berman, M. G., & Rosenberg, M. D. (2023). Functional brain connectivity predicts sleep duration in youth and adults. Human Brain Mapping, 44(18), 6293-6307.

Reviewer #2 (Kimberly Cote): The authors report on a multivariate data-driven analysis of the role of sleep quality and duration in biopsychosocial factors in a large sample of healthy young adults (n=770) from a publicly available dataset. Sleep and most outcome measures are based on self-report. Through canonical correlation analysis, the authors arrive at 5 latent components (LCs) that summarize the relationships between the subcomponents of the PSQI and 118 outcome measures. As a second aim, the authors describe the resting-state brain activity exhibited by the 5 LC profiles.

This is an interesting, data-rich, and well written paper offering some novel data. Examination of relationship between multiple variables in large samples is greatly needed to understand the complexity of the relationship between sleep and waking cognitive function and (mental) health. While self-report measures (particularly the PSQI for sleep quality) have some limitations, this is acknowledged by the authors, and the paper paves the way for future research to examine the LCs reported here using more objective measures of sleep. Additionally, while there are differences between objective and subjective measures of sleep, and the magnitude of this difference varies across individuals, there is still something important for health captured by the variable of one's perception of their sleep.

Sex differences are either controlled or investigated in the various analyses reported in the paper. For instance, it is reported that for LC5, higher sleep and biopsychosocial composite scores were found in females (page 7). More specific interpretation of the findings would be helpful. Is this consistent with the literature generally showing sleep is poorer and psychopathology more prevalent in females? Did the authors run the analyses in the females alone, and then the males alone? Are all 5 LCs seen within each sex? And does the strength of associations have a different pattern in males and females? For LC5, the authors explain that sleep disturbance may be due to a variety of factors fragmenting sleep, such as "difficultly maintaining optimal body temperature" (page 9). It would therefore be interesting in future research to see if this profile fits menopausal women in an older sample.

LC2 is a group with high psychopathology, but healthy sleep. The authors have interpreted this as possibly reflecting a resilient group whose sleep is not impacted by the stressors of psychopathology. This is an interesting individual difference hypothesis, one that could be investigated in future research to examine possible factors explaining the resilience. However, there are other possible explanations. This group could lack insight into their sleep disturbance, or not care enough to report it as a major concern (particularly relative to other concerns that may have more weight for them). Follow-up objective studies can address this. Another possibility is this group may not have developed the associated sleep disturbance yet - is there data in this sample to consider the length of time that symptoms of psychopathology have been problematic for individuals? It is possible that sleep disturbance may be present at a specific time, e.g., at the early onset of psychopathology (a vulnerability window).

It is mentioned that sleep disturbance in LC5 was associated with aggressive behavior (page 6, Figure 6). It would be informative to provide more information on the variable for aggression. Is it actually a behavioural task of aggression, or a self-report measure? Experimental research shows that sleep deprivation and restriction are associated with reduced aggressive behaviour on a laboratory task in men, a change that parallels reductions in testosterone following sleep loss.

Reviewer #3: Perrault et. al propose an original investigation of the multidimensionality of sleep and its links with the multidimension of socioeconomic and psychological factors on the one end (biopsychosocial factors - BPS) and brain function while awake one the other. They used data collected in >700 individuals as part of the Human connectome project with responses to a sleep questionnaire, many BPS variables and resting state fMRI recordings.

They decomposed the responses to a validated questionnaire in 7 dimensions that they related to >100 BPS features. They find that the first 5 (out of 7) latent components (LC) are "interpretable" are related to distinct set of BPS. They then report that these 5 latent components were significantly associated with distinct functional connectivity patterns.

While I salute the effort in proposing a novel view on sleep and the ambitious goal to make links between sleep BPS and brain function, I have several issues that leave me with a mixed impression of both positive and negative aspects. My main suggestion would be to remove a lot of the (unsure) results to leave more space to the stronger findings that could be covered in more details.

Main issue

The main strength and also my main issue is the fact that LC1 explains 88% of the variance between sleep questionnaire dimensions and BPS. This is in itself remarkable and in my view the most interesting result. One would think that if sleep and BPS are multidimensional then their links should be diverse as well. According to the results, it is not the case and this deserves in my view a larger emphasis.

The downside of the finding is that the other LCs only explain 4 to 1% of the variance. This raises questions about the relevance of these different LCs. Should the authors really focus on these LC? They should at least explain why it is interesting to focus on them and address the percentage of variance explained in their discussion.

They should also explain what makes LC2 to 5 interpretable and L6-7 not interpretable while they explain about 1% of the variance (similar to LC5). Considering the suppl. Fig. this feels unclear and rather subjective as I would see interpretation to both LCs.

In any case whether they detail 5 or 7 LCs means that they then relate 5 or 7 LCs to functional brain activity. Each LC is related to distinct sets of brain connectivity pattern. Because of the multiplicity of the findings the reader is overwhelmed in results which are not discussed in detail. In the end it is unclear why the functional connectivity results - acquired during wakefulness - tell us something important about the sleeping brain, the BPS and the subjective sleep dimensions.

The issue is in my view also very relevant because the statistical approach used by the authors, Canonical Correlation analysis (CCA) can be unstable depending on whether a dimension reduction is carried out or not (depending therefore on multicolinearity of the variables (see e.g. Yang et al. 2021 HBM). While it is likely not going to drastically change the profile of LC1 given its prominence, it may change the other LCs. This raises further concerns about the relevance of LC2-7 and warrants demonstration of the stability of the CCA analyses.

All this pleads in my view in rewriting the paper only including LC1 and its brain correlates (at least in the main text). This is in my view the results one can be most sure of and they are novel and interesting (though I leave the editor chose whether it is interesting enough for their journal)

Other important issues

- An issue that I would like to see addressed in the discussion is that CCA is inherently correlational, meaning that what composes an LC is driven by both sides of the correlation. The sleep questionnaire is therefore related to brain connectivity but also all the BPS associated to it in each LC. It is therefore at least plausible that the brain activity/connectivity recorded during wakefulness drives the BPS which are related to wakefulness in many instances which in turns is related to sleep dimensions.

- I salute the inclusion of many control analyses which shows the will of the authors to back up their findings using enough information so that it is clear to the reader. Reporting that changing the processing using CompCor instead of GSR leads to r~0.75 is, however, not convincing as it means that only 50% of the variance is common among both methods. The situation is even worse for when using sleep and biopsychosocial canonical scores instead of averaged scores where r~0.5. Also, I may have missed it but while the authors report that education level is related to at least 3 LCs (incl. LC1) they do not compute a control their analyses using education. Again, education may drive difference brain connectivity patterns and sleep dimensions.

- The main analyses include many subjects which were likely to have consumed psychoactive component prior to the fMRI acquisition. These were removed in a control analysis and it seems that it does not alter the findings. It would make much more sense to remove these individuals in the first place in a study investigation rs-fMRI.

- The fact that the sleep metrics, derived from a questionnaire, are inherently subjective should be further discussed as well. It is true that it makes the findings very practical/clinical (as mentioned by the authors) but does not tell us all about the underlying biology. Can the authors truly envisage an intervention/treatment based on their finding, shouldn't they need actimetry and/or electrophysiology to get to intervention targets?

- The manuscript makes no mention about time-of-day or prior sleep-wake history of the fMRI recording. Whether acquisitions were completed at the same time of day for each subject is not mentioned. One can imagine many reasons why BPS / Sleep quality could be related to chronic sleep restriction or to fMRI acquisitions being completed at a particular time of day. This could induce a systematic bias in the analyses which should ideally be controlled for and at minimum discussed.

Minor issues

- Introduction, second paragraph, 4th line: it is awkward to define sleep in terms of alertness. Do the authors mean daytime functioning?

- Figure 2 legend for panel (C) does not match with the text on page 13 under RSFC analysis where Fig2C is for uncorrected and 2D is for FDR-corrected network-wise beta coeff.

- Page 10, no reference is provided for "the U-shaped relationship of sleep duration with

- biopsychosocial measures. »

- The authors should consider discussing other multivariate papers such as Djonlagic et al. 2021 NHB which related objective measures of sleep to cognition and aging, which is not identical but somewhat similar to their study.

- Page 12, the authors should provide details about the missing data and data removed. At the moment we know what was done but not how much data was missing / removed.

---

## [Decision Letter · Decision Letter 2]

11 Jun 2025

Dear Dr Perrault,

Thank you for your patience while we considered your revised manuscript "A multidimensional investigation of sleep and biopsychosocial profiles with associated neural signatures" for publication as a Research Article at PLOS Biology. This revised version of your manuscript has been evaluated by the PLOS Biology editors, the Academic Editor and the original reviewers.

Based on the reviews and on our Academic Editor's assessment of your revision, we are likely to accept this manuscript for publication, provided you satisfactorily address the remaining points raised by the reviewers. Please also make sure to address the following data and other policy-related requests:

* We would like to suggest a different title to improve its accessibility for our broad audience: "Identification of five types of sleep-biopsychosocial profiles with specific neural signatures that link sleep variability with health, cognition and lifestyle factors"

* Please include the approval/license number of the ethical approval for the experiments.

* Please include information in the Methods section whether the study has been conducted according to the principles expressed in the Declaration of Helsinki.

* Please specify whether the participants provided written or oral consent.

* DATA POLICY:

Regardless of the method selected, please ensure that you provide the individual numerical values that underlie the summary data displayed in the following figure panels as they are essential for readers to assess your analysis and to reproduce it: 2A, 3A, 4A, 5A, 6A, S1, S4 and S7

* CODE POLICY

* If you have any references in the supplementary information, please move them to the main reference list.

* If you provide methodological details in the supplementary information, please move those to the main manuscript file as well.

We expect to receive your revised manuscript within two weeks.

*Published Peer Review History*

*Press*

Sincerely,

Christian

Christian Schnell, PhD,

Senior Editor

cschnell@plos.org

PLOS Biology

Reviewer remarks:

Reviewer #1: I thank the authors for their work on revising their manuscript based on my and other reviewers' feedback. There are a couple of minor changes that should be corrected:

1) "LC1 overwhelmingly explained 88% of the variance in a sample ..." is incorrect. Authors can estimate the explained variance via calculating out-of-sample r between the two sides (which is maximum going to be .69 based on the Figure 1, thus estimated R2 will be maximum 48%). The 88% is the proportion of covariance between sleep scores and BPS scores captures by the LC1 (not variance in the sample).

2) I would put a bigger caveat on LC3-5 as they are not only smaller in capturing covariance, but also seem to be not stable enough to cross-validate.

Reviewer #2: Comments to all reviews and revisions are appropriate.

Reviewer #3: The authors responded satisfactorily to most of my comments.

I still regret the limited discussion of the actual brain regions and/or networks in relation to each LC, and particularly to LC1. A lot fo discussion is about the what the different LCs represent in terms of BPS but much less what the brain region associated to the BPS/LCs tell us about sleep.

What does it tell us about sleep that the LC1 is associated with a negative connectivity between the TPN and DAN/SMN when the association is positive when considering TPB vs. DMN/CON. Likewise what do we learn about SubC or SAL connectivity (or absence of) considering LC1? Etc.

I realise that it is difficult to discuss all the associations uncovered given their large number. In my view, this makes the inclusion of the RSFC and of 5 (or 7) LC spectacular and/or technically very impressive rather than providing insights in the biology of sleep.

I would recommend increasing the discussion about the brain areas/networks by selecting some of the most striking connectivity findings, according to the authors (and beyond the segregation/integration ratio of the somatomotor and visual cortex). Else the RFSC feels rather technical in my view.

---

## [Editor Report · Decision Letter 3]

28 Aug 2025

Dear Aurore,

Thank you for your patience while we considered your revised manuscript "Identification of five sleep-biopsychosocial profiles with specific neural signatures linking sleep variability with health, cognition and lifestyle factors" for publication as a Research Article at PLOS Biology. This revised version of your manuscript has been evaluated by the PLOS Biology editors and the Academic Editor.

Based on our Academic Editor's assessment of your revision, we are likely to accept this manuscript for publication, provided you satisfactorily address the remaining data and other policy-related requests:

* There are a few references in the supplementary information that are not cited in the manuscript and listed in the main reference list. References only in the supplementary information will not be included in citation databases and give the authors the appropriate credit. Therefore, please include those references in the main manuscript file as well.

* You mention the S1 data file in which the source data are provided, but the file seems to be missing. Please include this when submitting the revised manuscript.

* Please note that GitHub repositories can be changed after publication. We therefore encourage you to archive this version of your publicly available GitHub code to Zenodo. Once you do this, it will generate a DOI number, which you will need to provide in the Data Accessibility Statement (you are welcome to also provide the GitHub access information). See the process for doing this here: https://docs.github.com/en/repositories/archiving-a-github-repository/referencing-and-citing-content

We expect to receive your revised manuscript within two weeks.

*Published Peer Review History*

*Press*

Sincerely,

Christian

Christian Schnell, PhD

Senior Editor

cschnell@plos.org

PLOS Biology

---

## [Editor Report · Decision Letter 4]

4 Sep 2025

Dear Aurore,

Thank you for the submission of your revised Research Article "Identification of five sleep-biopsychosocial profiles with specific neural signatures linking sleep variability with health, cognition and lifestyle factors" for publication in PLOS Biology. On behalf of my colleagues and the Academic Editor, Laura Lewis, I am pleased to say that we can in principle accept your manuscript for publication, provided you address any remaining formatting and reporting issues. These will be detailed in an email you should receive within 2-3 business days from our colleagues in the journal operations team; no action is required from you until then. Please note that we will not be able to formally accept your manuscript and schedule it for publication until you have completed any requested changes.

PRESS

Sincerely, 

Christian

Christian Schnell, PhD

Senior Editor

PLOS Biology

cschnell@plos.org